# Organisational kindness and compassion: what are the barriers, enablers and outcomes for clients and stakeholders?

Jennifer Smith-Merry[1], Damian Mellifont[2]*, Justin Newton Scanlan[3], Nicola Hancock[4]

1 Professor and ARC Industry Laureate Fellow, Centre for Disability Research and Policy, Faculty of Medicine and Health, The University of Sydney, Sydney, Australia, 2 Lived Experience Postdoctoral Fellow, Centre for Disability Research and Policy, Faculty of Medicine and Health, The University of Sydney, Sydney, Australia, 3 Associate Professor (Occupational Therapy), Centre for Disability Research and Policy, Faculty of Medicine and Health, The University of Sydney, Sydney, Australia, 4 Mental Health Stream Lead, Centre for Disability Research and Policy at The University of Sydney, Sydney, Australia

* damian.mellifont@sydney.edu.au

## Abstract

Unkind bureaucratic policies such as the Australian Robodebt policy, which targeted welfare recipients with automatic debt letters, and are geared towards economic savings, can significantly harm those impacted by them. Compassion and kindness are receiving increased research attention related to how organisations work internally. However, a greater investment in studies is needed to increase understandings about how compassion and kindness can underpin interactions with external stakeholders. Addressing this research gap, we aimed to identify barriers, enablers and outcomes to organisational kindness and compassion informed by the literature, and to propose future research directions related to organisational kindness and compassion towards external stakeholders. A search of four scholarly databases identified 25 relevant publications. Thematic analysis of included publications revealed barriers of commodification, personal risks, dysfunctional environments, inauthentic attempts at, and a lack of understanding of the need to be compassionate or kind. Enablers included building compassion into organisational policies, processes, practices and activities, compassion contagion, training of staff, leading with compassion, and kind and compassionate communication. Outcomes of kindness included building positive and healthy relationships with stakeholders, supporting positive experiences among stakeholders, and contributing to an organisation's profitability, productivity, performance and standing in the community. We conclude by recognising that kindness is essential for ongoing trust in health and social care institutions and government policy.

**Data availability statement:** All relevant data are within the paper and its Supporting information files.

**Funding:** The author(s) received no specific funding for this work.

**Competing interests:** The authors have declared that no competing interests exist.

## Introduction

Kindness is defined as "not treating someone well, or not considering someone's feelings" [1]. Significant problems arise when organisations, including those in government bureaucracies, are unkind to stakeholders in the way that they conduct their mission and implement services. Positioning social welfare as a focus area of this paper and commencing with an Australian policy example, unkindness is clearly evidenced in the former Australian Government's now abandoned 'Robodebt' scheme introduced in 2016. Under Strengthening the Integrity of Welfare Payments policy [2], the scheme, an AI based algorithm compelled Australian social security clients identified by the algorithm to prove they had not made illegitimate unemployment or disability benefits claims (as far back as seven years prior) [3]. The main objective of the scheme was to achieve budget savings [4,5]. The unkindness of this welfare policy was captured in the findings of the Robodebt Royal Commission. The Commission used a gamut of damning terms to describe the scheme's operation, including 'cruel', 'reprehensible' and 'illegal' [6]. Prior to and during its operation public service administrators, policy staff and lawyers had all expressed serious doubts about the scheme's legality but feared pressing the matter further with supervisors leading to significant consequences for those people subject to it [7]. This was clearly an unkind policy, implemented by a bureaucracy which had dehumanised the population of welfare recipients it was designed to serve, and where individuals working within it felt unable to act even in the face of strong evidence of negative outcomes to recipients. By requiring vulnerable welfare recipients to repay government issued false debt notices, the Robotdebt scheme resulted in humiliation, distress and at least three cases of death by suicide [8].

While extreme in its impact, this is not an isolated example with other unkind, process-oriented care and support systems negatively impacting upon people who receive support. Australia's National Disability Insurance Scheme (NDIS) is described as one of the most significant social policy reforms that the nation has seen since the introduction of Medicare [9]. NDIS stakeholders, however, have reported unkind aspects of the Scheme. The Scheme is critiqued, for example, for its confusing and burdensome administrative processes, a lack of accommodations in place to allow effective communications between participants and staff, and a dismissal of participant concerns or preferences in NDIS planning activities [10,11]. These are inherently unkind processes because they do not consider the individual and their needs, but rather prioritise structured bureaucratic processes. Despite "choice and control" for participants in relation to their supports being central tenets of the NDIS, bureaucratic processes that do not consider individual needs can actually diminish choice and control for NDIS participants [12]. A perceived lack of concern for participants and their needs, in favour of Scheme processes and rules, has also led to a broader lack of trust in the National Disability Insurance Agency (NDIA), which is the organisation administering the NDIS. This means that people are more likely to fear any interactions with, or decisions made by, the NDIA and question the motivation for decisions [13,14]. Unkindness therefore undermines the effectiveness of bureaucratic organisations in conducting their social purpose and legislatively mandated service roles.

Anderson and Brownlie [15, p.5] define kindness as, "the things that people do for one another (both practically and emotionally) in response to moments of perceived need, when there is the option to do nothing." Kindness has been critiqued as a weak concept that can deflect attention away from injustice [16,17]. Nevertheless, there has been a recent recognition of the importance of kindness in addressing the disconnect felt by many individuals from their communities and the social institutions that should be there to support them. Rowland [18] purported that kindness can hold little ground in a world dominated by individualism, where an individual is driven to compete to gain advantage over others. However, following the arrival of Covid-19 and pandemic related uncertainty, the importance of kindness, compassion and connections among individuals and organisations has been highlighted [19,20]. In this challenging context, the kindness of organisations was related to increases in wellbeing by fostering social connections [21,22]. The concept of kindness is also related to the reduction of loneliness and isolation as significant policy challenges, together with growing inequalities and social divisions [23]. It is therefore important for policymakers to have a lens that allows for the application of kindness to advance the social and economic wellbeing of individuals and communities [23].

*Organisational kindness* is both the antithesis of the negative aspects of 'unkind' organisations, described above, but also describes an organisation whose structures enable the humanising of individuals through prioritising key behaviours and actions which characterise what it is to be 'kind'. Stepping back from bureaucracies and institutions, to thinking more generally about the term, we find that kindness is associated with both thoughts and actions. For example, Canter, Youngs and Yaneva [24] argue that the three underlying components of kindness are: a) tolerance for those around us; b) empathy for the feelings of others, and c) principled actions through fair behaviours. It is also primarily considered in relation to others. Kindness tends to be thought of as action underpinned by a motivation to assist another person rather than a motivation to gain an overt reward or to avoid punishment of some kind [25,26]. Kindness thus consists of behaviour that is conducted in a thoughtful way (i.e., the act of kindness) [27]. From an ethical perspective, kindness is characterised by a deep and genuine care for other people [28,29]. Kindness can be understood in terms of holding respectful interest in and compassionate understandings and actions towards others [21,30] and within an organisational context this is translated within the processes and practices of organisations, both administrative and interpersonal. Kindness can also be a direct function of policy. For example, when incorporated within regulatory policies and processes in health, kindness can provide a framework which purposefully focuses on patient safety while also considering the wellbeing of healthcare practitioners [31].

Kindness is frequently linked to and applied interchangeably with other terms including that of compassion [32–34]. Compassion is depicted as the emotional undercurrent of kindness [28,35] and can be considered both as a component of kindness and as a discrete process. Compassion is one of the six fundamental manifestations of kindness with the others consisting of empathy, respect, fairness, altruism and care [21,36]. Moral duty, sentimentality and conscience are considered to be drivers of compassion [37]. Compassion, manifests in acknowledgement, empathy, and reaction, and is focused upon practically redressing externally situated pain [38,39]. This is important within organisational interactions with individuals, where difficult decisions must sometimes be made (e.g., refusing access to public services) which can cause individuals pain. Compassion incorporates a response or action component [33,40–42] and is present on occasions where people are aware of and respond with kindness to the pain experienced by others [43]. As applied in organisational settings, kindness and compassion can help to build and sustain trusting partnerships with external stakeholders by holding open conversations about stakeholders' concerns [44,45].

Kindness and compassion have been measured in both qualitative and quantitative ways including quantitative self-reporting measures involving the use of questionnaires and surveys and also via qualitative interviews and focus groups [46–48]. Research has indicated that while the everyday giving and receiving of kindness might be construed as mundane, these acts hold tremendous emotional relevance to the individuals involved in kind interactions as both 'giver' and 'receiver' [49–51]. Individuals who practise kindness experience benefits of happiness and satisfaction [34,52]. Furthermore, kindness can offer physical benefits. A simple act of kindness can raise serotonin and improve immune system

functioning in both the giver and receiver [53,54]. Compassion too is associated with benefits of inner wellness and insight for the giver [55,56]. Receivers of kindness report increased wellbeing in the short-term (i.e., within a week) and increased happiness over the long-term (i.e., after two months) [57].

Organisational compassion involves employees collectively acknowledging, feeling, and responding to the distress experienced by others [58,59]. Organisational compassion has a long history, having gained traction as a concept during the Great Depression and its economic challenges where unemployment was widely experienced [60,61]. It was not until the 1970s, however, before organisational compassion was explicitly referenced in a text entitled, *Without sympathy or enthusiasm: The problem of administrative compassion* where economic imperatives were described as constraining organisational compassion [60,62]. However, a ruthless focus on achieving stakeholder wealth at any cost is facing increasing public scrutiny, providing a shift of kindness into the organisational sphere, together with opportunities for organisational leaders to have serious conversations about kindness [63–65].

Contemporary organisational kindness and compassion, however, should not always be assumed to be authentic. Shared construction of organisational kindness is an ongoing, social and reflective process which results in a range of practices [21]. This includes workshops, leadership modelling guides, and media messaging that encourages positive interactions between organisations employees and external stakeholders [66,67]. Compassion, as enacted in organisational settings, can range from a 'tick a box' approach to comply with organisational standards and inauthentic stakeholder relationships, all the way through to compassionate acts that are spontaneous and sincere [37,60]. As applied throughout this paper, the term 'inauthentic' is defined as "not real, true or what people say it is" [68]. Beyond organisational settings to those in leadership positions, we recognise that political leaders who provide their public support for kind and compassionate policies, are often seen as inauthentic or weak, and take on a level of political risk. New Zealand's former Prime Minister Jacinda Ardern, for example, received much criticism, particularly among male constituents, who perceived her decision to use maternity leave (itself, an example of compassionate policy) as a sign of weakness [69].

The topics of kindness and compassion have gained momentum as an important and productive area of research inquiry in relation to those working within organisations [70,71]. That research has primarily investigated the fostering of manager employee relationships in the technology sector and more broadly. However, there is a significant research gap related to understanding the nature and impact of organisational kindness and compassion in relation to external organisational stakeholders [44,45]. In this context, external organisational stakeholders are the customers, clients, recipients of support, or investors that are external to an organisation and who are affected by organisational activities, as opposed to internal stakeholders, such as staff, who are impacted by organisational activities in different ways [72,73]. They are impacted by the decision making of organisations in a way that is distinct from those internal to the organisation. Aiming to contribute to understanding how organisational kindness manifests in this setting, we conducted a scoping review to: a) critically investigate what the literature can tell us about organisational kindness and compassion in terms of respective barriers, enablers, and outcomes for clients and other external stakeholders; and b) identify research opportunities to improve understanding about organisational kindness and compassion for external organisational stakeholders. While kindness and compassion come from individuals within organisations who create organisational systems and processes and the enactment of those processes [28,38], we are intentionally reifying organisations as places where the enactment of kindness resides. Responding to Australia's Robodebt policy scandal and persistent NDIS shortfalls, our study concludes that organisational kindness and compassion can deliver positive outcomes for external stakeholders.

## Method

This study has been guided by the Arksey and O'Malley [74] described stages to conducting a scoping review. These stages consisted of: 1) setting the research direction; 2) identifying potentially relevant studies; 3) selecting relevant studies; 4) extracting data from relevant studies; and 5) summarising the study findings. The practical application of each

of these five stages is described as follows. Researcher initials (i.e., JSM, DM, JNS) are used throughout to identify which researcher conducted which research-related task.

*Stage 1 – setting the research direction.*

To set the research direction, the authors collaboratively developed the research aims. The research aim was refined into the following research question: as applied to external stakeholders, what are the barriers, enablers and outcomes of organisational kindness and compassion?

*Stage 2 – identifying potentially relevant studies.*

The search terms (see Box 1) were collaboratively developed among the paper authors. Potentially relevant studies were then identified by applying the search terms to the following databases: Scopus; ProQuest Central; PsycINFO; Business Source Ultimate via EBSCO. These databases were chosen because they would provide papers relevant to understanding organisational compassion or kindness. DM exported references for the potentially relevant studies into EndNote before importing these into Covidence where duplicates were removed ready for two-reviewer screenings. The most recent search was conducted on 25 October 2023.

---

### Box 1. Search terms applied to scholarly database.

"corporate kindness" OR "organi* kindness" OR "bureaucratic kindness" OR "radical kindness" OR "infrastructure of kindness" OR "corporate compassion" OR "organi* compassion" OR "kind* organi*" OR "kindness innovation" OR "kindness in leadership"

fields = all fields.

---

*Stage 3 – selecting relevant studies.*

Relevant records were selected through the application of the inclusion and exclusion criteria as identified in Box 2. A ten-year period for this scoping review was purposefully chosen by the authors for its capacity to encompass scholarly insights about the barriers and enablers of organisational kindness and compassion for external stakeholders that are contemporary with current policy and organisational practices. DM and JSM conducted the abstract and full text screenings with any conflicts resolved by a third reviewer (JNS). DM imported the final collection of included publications into NVivo for qualitative analysis. NVivo is a comprehensive software package used to manage and share the data and analyses between the research team members throughout the qualitative analysis of data [75].

---

### Box 2. Inclusion and exclusion criteria.

Inclusion criteria: a) document type = journal article, scholarly report, thesis (PhD or Masters) or book chapter; AND b) publication year = 2013–2023; c) document informs about public or private sector organisations in terms of: enablers or barriers of kindness/compassion; and/or kindness/compassion related outcomes.

Exclusion criteria: a) language is not English; b) full document is not available; c) document speaks about internal organisational kindness/compassion practices and not impact on those outside of the organisation (e.g., universities to students or health care organisations to patients).

---

*Stage 4 – extracting data from relevant studies.*

A charting framework was collaboratively developed among authors. As recommended by Arksey and O'Malley [74] this framework lists the information fields that the research team agreed to extract from included publications. These agreed fields included: record number; year; authors; title; publication type (research article, report, thesis, book chapter); research methodology; study location; study limitations; how kindness/compassion is described; kindness/compassion context (organisation description); external recipients of kindness/compassion; and content related to barriers, enablers and outcomes of organisational kindness and compassion to external stakeholders. Using this framework, DM extracted data from included studies into a spreadsheet.

*Stage 5 – summarising the study findings.*

We conducted an inductive 'data driven' thematic analysis, which did not seek to follow an existing theoretical framework, but derived results from the themes which emerged from our open coding of the data [69]. Thematic analysis is an iterative process of identifying themes, naming themes, and revising themes [76]. Applying this approach, DM inductively coded the 25 records. Following team reviews and deliberative discussions, consensus about the thematic coding was reached.

## Results

We identified 489 potentially relevant publications from the scholarly database searches (see Supplementary File 1). Of these, we identified a total of 25 publications that met our inclusion criteria (see Fig 1 for PRISMA diagram). Included publications consisted of 15 research articles, 9 book chapters and one report. Almost all included publications were qualitative studies (n = 24) primarily based on interviews. Analysis involved: thematic analysis (n = 12); interpretive analysis (n = 9); qualitative modelling (n = 2) and grounded theoretical analysis (n = 1). One quantitative study utilised a survey. Table 1 provides content details of the included publications (i.e., publication type and method applied) as well as the excluded studies. The themes are conceptually displayed in Fig 2 and divided into barriers and enablers and resulting outcomes and the concepts that make up each. The following results section further explores each of these themes.

### Barriers to organisational kindness and compassion for external stakeholders

Organisational leaders were viewed as integral to promoting kindness within their organisations. This related to actions that they took to lead their organisations, rather than individual actions taken external to those roles (e.g., personal giving). Seven publications reported on the potential *personal risks for organisational leaders who promote compassion or kindness to external stakeholders*. For example, Haskins and Thomas [90] noted that some managers in universities associate kindness with a lack of resolve and weakness of character. In their study informed by interviews with UK based private sector (n = 15) and public sector (n = 15) organisational leaders, Murray and Gill [89] also recognised kindness being widely misconceived as weakness among employees of private organisations. For those engaged in promoting kindness in their organisations, the literature cautioned of kindness burnout and compassion fatigue as leaders become overwhelmed and exhausted with their concerns for external stakeholders [19,27,40,82,83].

*Dysfunctional, toxic and stressful organisational environments* as barriers to implementing kindness in organisations were reported in six publications [e.g., 86, 126]. The literature cautioned that people who are treated badly in organisations can in turn treat others unkindly [80,88]. Belak and Waddington [87], for example, cautioned about the potential for academics to bully students in a dysfunctional and toxic higher education setting. Stressful settings were also described as producing compassion fatigue among healthcare organisations which could negatively impact upon the patient care that is received [81].

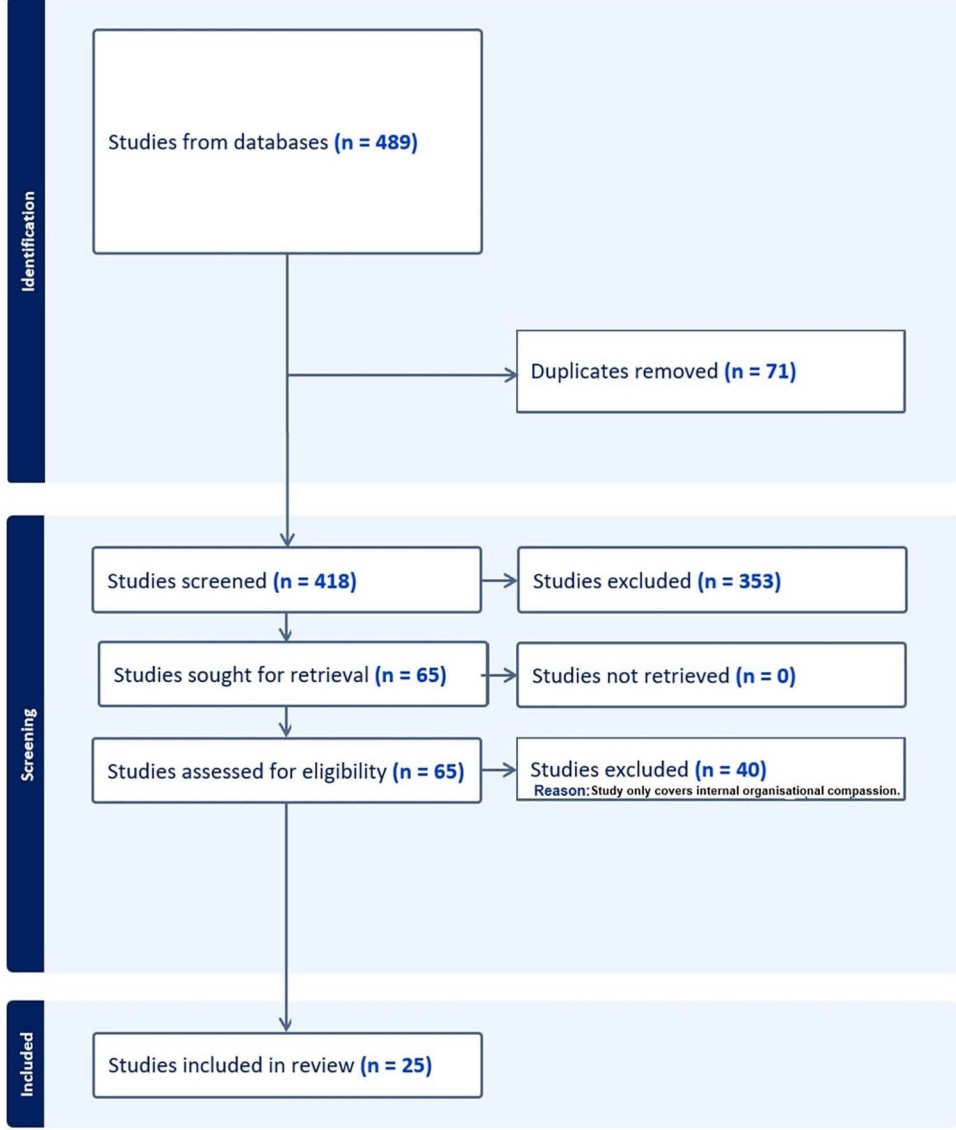

**Fig 1. PRISMA diagram.**

Six publications described *inauthentic attempts at showing compassion or kindness to external stakeholders* [e.g., 40, 45]. Having surveyed faculty, student and non-teaching staff (n = 225) at a Saudi Arabian higher education institution, Al Kahtani, Faridi and Kuchar [44] cautioned about faculties compromising students' learning by talking about compassion but failing to act in compassionate ways. Similarly, in their study investigating organisational compassion and informed by professional representatives from 32 Portuguese organisations, Araújo, Simpson, Marujo and Lopes [37] warned of organisations who alienate external stakeholders through their inauthentic attempts to show compassion. The literature also cautioned against tokenistic displays of organisational compassion to communities in crises where receipt of public recognition and praise is prioritised over helping people in need [42,82].

**Table 1. Included publications, content details and excluded studies.**

**Publication type (N = 25)**

Research article (n = 15): Shillington, Morrow, Meadows, Labadie, Tran, Raza, Qi, Vranckx, Bhalla and Bluth [25], Araújo, Simpson, Marujo and Lopes [37], Grover [40], Simpson, Clegg and Freeder [42], Al Kahtani, Faridi and Kuchar [44], Forester [45], Villiers [64], Simpson, Farr-Wharton and Reddy [77], D'Souza and Martí [78], Espedal [79], Friedman and Gerstein [80], Smith-Macdonald, Venturato, Hunter, Kaasalainen, Sussman, McCleary, Thompson, Wickson-Griffiths and Sinclair [81], Surman, Kelemen and Rumens [82], Vogus and McClelland [83], Vogus, McClelland, Lee, McFadden and Hu [84].

Book chapter (n = 9): Lawrence [19], Haskins, Thomas, Bennett, Gibb, Gibb, Gill, Johri, Murray and Rowland [27], Farquharson [34], Haskins [85], Matthewman [86], Belak and Waddington [87], Gibb, Gibb and Bennett [88], Murray and Gill [89,90].

Report (n = 1): Mills, Thom, Black and Quince [16].

**Method applied (N = 25)**

Thematic analysis (n = 12): Mills, Thom, Black and Quince [16], Lawrence [19], Shillington, Morrow, Meadows, Labadie, Tran, Raza, Qi, Vranckx, Bhalla and Bluth [25], Araújo, Simpson, Marujo and Lopes [37], Grover [40], Simpson, Clegg and Freeder [42], Espedal [79], Smith-Macdonald, Venturato, Hunter, Kaasalainen, Sussman, McCleary, Thompson, Wickson-Griffiths and Sinclair [81], Surman, Kelemen and Rumens [82], Haskins [85], Gibb, Gibb and Bennett [88], Murray and Gill [89].

Interpretive study (n = 9): Haskins, Thomas, Bennett, Gibb, Gibb, Gill, Johri, Murray and Rowland [27], Farquharson [34], Forester [45], Villiers [64], Friedman and Gerstein [80], Vogus and McClelland [83], Matthewman [86], Belak and Waddington [87], Haskins and Thomas [90].

Qualitative modelling (n = 2): Simpson, Farr-Wharton and Reddy [77], Vogus, McClelland, Lee, McFadden and Hu [84].

Grounded theory (n = 1): D'Souza and Martí [78].

Quantitative statistical analysis (n = 1): Al Kahtani, Faridi and Kuchar [44].

**Excluded studies (N = 40)**

Paakkanen, Martela, Hakanen, Uusitalo and Pessi [91], Liu, Luo and Tang [92], Thomas and Rowland [93], Simpson and Berti [94], Kasekende, Nasiima and Byamukama [95], Simpson, Farr-Wharton, e Cunha and Reddy [96], Tsui [97], Fry [98], Quinane, Bardoel and Pervan [99], Meyer [100], Thomas and Rowland [101], Denney [102], Kanov, Powley and Walshe [103], Guo and Wang [104], Hart and Hart [28], Simpson, Rego, Berti, Clegg and Cunha [105], Madden, Madden and Smith [106], Thienprayoon [107], Brandert and Matkin [108], Bolino and Grant [109], Araújo, Marujo, Lopes and Pereira [110], Simpson, Clegg and Pina e Cunha [111], Datu, Buenconsejo, Valdez and Tang [22], Haskins and Thomas [112], Simpson, Clegg, Lopes, e Cunha, Rego and Pitsis [113], Ahmad, Islam, D'Cruz and Noronha [114], Chatterjee, Chakraborty, Fulk and Sarker [115], Thomas and Rowland [93], Onken-Menke, Lauritzen, Nuesch and Foege [116], Tietsort, Tracy and Adame [117], Simpson, Clegg and Pitsis [56], Lee [118], Haskins [119], Williams and Shepherd [120], Simpson, Clegg and Pitsis [60], Simpson, Pina E Cunha and Rego [121], Shahzad and Muller [122], Nazir and Islam [123], Johri [124], Simpson, Cunha and Clegg [125].

Reason for exclusion: document speaks about internal organisational kindness/compassion practices and not impact on those outside of the organisation (N = 40).

Six publications described *commodification that places profits over compassion or kindness for external stakeholders* as being barriers to organisational kindness [25,45,81,119]. Business focus and related stress and time pressures were described as worsening the attention received by external stakeholders including healthcare patients and students in higher education [34,126]. This lack of attention meant that poorly treated patients experienced pain and students experienced damaged relationships with academic staff [34,126].

Three publications raised the barrier of *organisations having problems understanding a need to be compassionate or kind*. Two papers recognised that while it is easier to understand medical staff showing kindness to patients, there was less evidence and understanding about kindness and compassion shown to stakeholders of private sector organisations [27,64]. Belak and Waddington [87] reported on the challenge to understand what student compassion means in neoliberal higher education that delivers education in an impersonal manner. Authors cautioned that compassion towards and understanding of external stakeholders' needs can be undermined by the self-interest of private companies [64], and within the growing neoliberal higher education sector focused on economic goals [87].

### Enablers to organisational kindness and compassion for external stakeholders

Eight publications discussed a capacity to *build compassion and kindness for external stakeholders into organisational policies, processes and practices* as processes enabling kindness in organisations [16, 81, e.g., 126]. The tone of kind and compassionate organisational policy and practice reflects principles of justice, respect and empathy for stakeholders

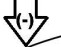

**Barriers**

* personal risks for promoting k & c to external stakeholders
* dysfunctional, toxic and stressful organisational environments
* inauthenticity at showing k & c to external stakeholders
* placing profits over k & c for external stakeholders
* problems understanding a need to show k & c.

**Enablers**

* k & c built into policies, processes and practices
* k & c expressed via organisational activities
* compassion among staff spreading to external stakeholders
* k & c to external stakeholders encouraged through training
* k & c in communicating with external stakeholders
* leading with k & c towards external stakeholders

(-)          (+)

**Outcomes**

* building positive relationships via k & c
* k & c contributing to organisational profitability, productivity, performance and standing
* k & c enabling positive experiences for external stakeholders

**Fig 2. Kindness and compassion – the enablers, barriers and outcomes.** Notes. (-) = hindering outcomes; (+) = supporting outcomes; K&C = kindness and compassion.

[19,89,90]. Respect is defined as "admiration felt or shown for someone or something that you believe has good ideas or qualities" [127]. In their exploration of the concept of organisational compassion in a healthcare context and supporting compassionate actions of individuals (i.e., staff), Simpson, Farr-Wharton and Reddy [128] called for organisations to explicitly incorporate compassionate and caring practices (e.g., showing respect for patients) as a goal for staff within new role descriptions. Aiming to describe practical and evidence-based ways of leading organisations with a kind heart, Farquharson [34] reported that compassion and kindness to university students can be supported by staff who behave professionally by showing respect for students as people.

Seven publications described *conducting organisational activities that express compassion and kindness towards external stakeholders* [e.g., 83, 85, 90, 129]. Activities discussed across papers included performing community support activities (e.g., providing gifts to vulnerable people) [37] and the distributing of stress management resources to students by 'compassionate coordinators' who report directly to the head of a university [44]. Shillington, Morrow, Meadows, Labadie, Tran, Raza, Qi, Vranckx, Bhalla and Bluth [25] also described a range of kindness activities in universities including developing a kindness checklist for faculty to support their students, building kindness within course design (e.g., lectures), and promoting kindness on campus through infographics and posters.

Seven publications reported on *experiences of compassion among colleagues which in turn encourages compassion for external stakeholders* [e.g., 40, 42, 128]. For example, Simpson, Farr-Wharton and Reddy [128] described the potential of individuals, in this case frontline healthcare workers, who experience compassion from colleagues to pass this compassion onwards in efforts to reduce the suffering of others. Several papers described organisational compassion for customers and clients as having a contagion effect [19,80,89]. In their discussion of kindness in leadership roles across UK private and public sector organisations, Murray and Gill [89] reported on kindness shown among public service colleagues 'spreading' to services users (e.g., patients). Lawrence [19] identified the possible start of a compassion contagion following a pandemic which encouraged academic staff to show kindness and compassion towards university students by prioritising their health and safety. These papers all point to the contagion effect of compassion within organisations spreading to the external stakeholders and clients of those organisations.

Six publications discussed the *training of staff to show compassion or kindness to external stakeholders* [e.g., 44, 87]. For example, empathy training programs, developed to practically demonstrate the importance that the medical profession has placed on compassion and kindness, were used to encourage individuals (i.e., hospital staff) to be compassionate and kind to patients [90]. The literature also recognised the capacity of compassion-focused leadership training programs to widely promote compassion and kindness towards service users (e.g., students, patients) [27,86,89].

Four publications referenced *communicating in compassionate or kind ways with external stakeholders* as being an important facilitator of kindness [e.g., 64, 87]. For example, Araújo, Simpson, Marujo and Lopes [37] discussed organisational compassion in terms of bringing external stakeholders into open and respectful conversations to enable their needs to be accommodated. Mills, Thom, Black and Quince [16] also stressed a need to promote kindness through active listening in the support of court defendants.

Four publications reported on *leading with compassion or kindness towards external stakeholders* [e.g., 64, 126]. These papers collectively characterised leadership for kindness as flexible and considered leadership at any level of the organisation. Investigating attributes that align with a compassionate university, Belak and Waddington [87] highlighted the need for compassionate and emotionally intelligent leadership in universities to redress the bullying and harassment of students by openly speaking out against such behaviours. In their qualitative study informed by 19 UK based food banks, Surman, Kelemen and Rumens [82] reported instances where individuals (i.e., volunteers) who lead with kindness and position people as a priority through the bending of rules and recognising and meeting the immediate physical needs of food bank users.

## Outcomes of displays of kindness and compassion for clients and other external stakeholders

Seven publications reported on the importance of the process of *building positive and healthy relationships with stakeholders or clients through kindness and compassion* [e.g., 27, 89]. Two papers focusing on higher education discussed compassionate teaching practices, which included holistic teaching approaches involving mind, body and spirit [86] and valuing students' individuality and wellbeing [90]. These practices were shown to improve relationships with university students who are treated with care, consideration and honesty [86,90]. The literature also noted that 'compassionate care', described as responsive and personalised healthcare interactions, supports meaningful relationships with patients and their families [81,84]. Organisational kindness was reported as delivering strong and trusting relationships with external stakeholders (e.g., defendants in specialist courts and patients) through relaxed courtroom procedures where jokes, laughter and applause are allowed [16], and patient empathy education for hospital staff [90].

Six publications discussed compassion or kindness towards external stakeholders as *contributing to an organisation's profitability, productivity, performance, and their standing in the community* [e.g., 37]. Kindness and compassion within organisations and workplaces improved customer satisfaction and loyalty, enhanced sales, and lowered marketing costs [34,80,119]. Haskins and Thomas [90, p.44] reported customers' appreciation for organisations who are willing to "go the extra mile" as these organisations are the ones who stand out from the crowd. Organisational kindness was also described as generating community goodwill through the employment of people from local communities and engaging in local activities [88,90].

Five publications described *compassion and kindness enabling positive experiences for organisation stakeholders and clients* [e.g., 64]. These experiences were discussed in relation to higher education and healthcare settings. In particular, compassionate dealings with traumatised students in academic settings were seen to increase recovery and lower drop-out rates [44]. Shillington, Morrow, Meadows, Labadie, Tran, Raza, Qi, Vranckx, Bhalla and Bluth [25] also noted kindness as a central part of positive student experiences in higher education by promoting their belonging. Compassionate, person-centred healthcare that meets whole-of-person needs was also said to deliver improvements in health outcomes [83], and could assist in recognising and mitigating patients' experiences of pain [126].

## Discussion

Addressing the first of our research aims, our scoping review has identified positive outcomes for organisations and their external stakeholders when organisations invest in compassionate and kind policies and practices. It has also revealed a diversity of barriers and enablers which respectively constrain and advance organisational kindness and compassion for external stakeholders. Furthermore, our findings provide organisations with an initial understanding of those practices which serve as barriers or facilitators to enacting organisational kindness.

Informing the second of our research aims and explaining why healthcare and education dominate the findings, the majority of the existing literature related to healthcare and educational organisations. A paucity of literature on organisational kindness beyond healthcare and education policy settings, in other social care bureaucracies and organisations is therefore a significant finding of our study. This gap is notable in the face of growing interest in kindness as a social value and public concern about unkind organisations and bureaucracies in a broad range of settings [e.g., 130–132]. In healthcare, kindness and compassion offered improved relationships *with* healthcare clients and improved health outcomes *for* healthcare clients [81,83,84]. Aligning with organisation behaviour framework's micro (i.e., individual level) analysis, these desirable outcomes were enabled through *individualised*, responsive, and compassionate care, and also via compassionate communication with health service users [81,84,133]. Findings from health care organisations indicate that thinking about compassion and kindness is more advanced or centred in these settings, perhaps because this has aligned with a growing focus on patient-centred care [134]. Hence studies are required to examine possible opportunities to advance this micro-level focus as prominent in the health care sector to better support stakeholders with disability in the welfare sector. There are also some initial shared findings from the review which can point the way to further research that should take place. These research directions might encourage deeper investigations to connect kindness with desirable outcomes for external organisational stakeholders (e.g., service users and higher education students). This includes the advancement of anti-racism in social service organisations through kindness training of service staff [132] and an interruption of whiteness in students' learning through a decolonizing approach to higher education which supports the inclusion of Indigenous social work educators [135].

To dismiss concerns and instead embrace unkind bureaucratic processes and programs is to put at risk public trust in government welfare supports. Our paper began by highlighting Australia's Robodebt policy scandal which drew attention to a loss of kindness and compassion towards social welfare clients through stressful and dysfunctional public organisational settings [89] and the unkind business logistics of government bureaucracy [126]. The Royal Commission into the Robodebt Scheme findings [136] showed that where kindness and compassion are absent in policymaking it can lead to very negative outcomes, in that case, the loss of lives through suicide. Kindness applied as a policy instrument has a role in addressing the lack of compassion shown towards external stakeholders (i.e., issues identification and options analysis stages) and the failure for action once the negative impacts of unkind organisations are known (i.e., implementation and evaluation stages). Systems theory endeavours to improve understandings about systems and their practical implications [137]. By positioning external stakeholders as a key component of welfare systems, a systems theory approach can also help us to think through how compassion is supported within systems more broadly, instead of just within the individual organisation. In terms of reforms towards more compassionate policy, the Australian Government has agreed to accept in full 49 of the 57 recommendations from the Robodebt Royal Commission. By implementing recommendations including considering of external stakeholders' vulnerabilities, providing policy oversight, and designing policies with a focus on the people they are meant to assist means that in practice, this policy debacle will not repeat itself [138,139].

The findings from our study support the potential for a kindness contagion effect whereby the internal operating environment of organisations can lead to the expression of kindness and compassion to external stakeholders. Public servants who treat each other with kindness and compassion are in turn encouraged to interact with external stakeholders in similar ways [89]. By embracing compassion in organisational policies and practices to address suffering among

individuals in communities [47,59], compassion can become the normalised operating environment projected both within and beyond the organisation. We call on Australian public service leaders to promote progressive work environments where respect and kindness for government colleagues and external stakeholders are valued. This valuing of organisational kindness and compassion opposes toxic workplace settings which silence objections to illegal activities that may be encouraged by the government of the day. Crucially, embedding kindness into policymaking activities positions social justice for external stakeholders as a pillar of good public administration.

Positive interactions with organisations, including government bureaucracies are also important for creating an environment of trust between the organisation and its stakeholders or clients. A lack of trust can mean that every interaction is viewed as a threat and continual mistrust erodes public support for public institutions, as demonstrated in the example of the Australian National Disability Insurance Agency [13,14]. A decline of trust undermines the legitimacy of public service organisations and the decisions that they make [140]. Trust is a multi-layered concept, but there is a role for interpersonal and institutional/structural interactions to enable trust and kindness to be positioned as a underpinning concept for such interactions [141]. However, these propositions about the place of kindness in building trust are not tested and our research showed a lack of existing research to specifically consider kindness and compassion in relation to public service bureaucracies more generally. Investment is therefore required in studies which investigate: a) the ways that kindness can be promoted within the internal and external public service organisations b) the impact of such practices on external stakeholders and clients and their trust in these organisations.

Our scoping review identified a perception of personal risks to leaders as a barrier to kindness and compassion. Former New Zealand Prime Minister Jacinda Ardern's leadership faced criticism, but negative responses to compassionate leadership are not inevitable [69]. Ardern was publicly praised for integrating leadership and compassion through health policy messages in which external stakeholders (i.e., constituents) were encouraged to adhere to Covid lockdowns [105,142]. Ardern's decisive, pragmatic, and compassionate leadership achieved highly desirable outcomes of saving lives in the first instance, and saving livelihoods in the second, by allowing the New Zealand economy to safely open sooner [105]. In the face of criticism, leading with kindness requires a strength to move beyond worrying about personal attacks of character and to be resolute on achieving compassion-driven outcomes. However, this is not a given in leaders and requires organisational practices which sit around them to support action. Understanding of the characteristics of leaders and their organisations which promote compassionate leadership was a gap in the existing literature. We therefore encourage future research which investigates the degree to which showing compassion and kindness to clients and other organisational external stakeholders can be effectively taught through leadership training programs, and which organisational structures best promote compassionate leadership for building kind bureaucracies.

## Limitations

Our scoping review has provided an important contribution to knowledge about kindness and compassion involving external organisational stakeholders. Our findings are purposefully limited to the search terms applied and databases accessed, these findings may potentially be expanded through a broader focus on kindness and compassion. While a finding rather than a limitation of our own study, the findings are primarily based on qualitative studies with small sample sizes, raising concerns about generalisability. Additionally, the possibility of selection bias, due to non-random participant selection, should be acknowledged. There is therefore a need to further test research findings through additional empirical investigations [42,81,84,126,129].

## Conclusion

These findings offer a framework for understanding how organisational compassion and kindness influence external stakeholders, as well as the factors that enable or hinder these practices. Our review revealed a multitude of compassion and kindness barriers involving external stakeholders that include commodification of business purposes, personal

risks, dysfunctional environments, inauthentic attempts, and a lack of understanding of the need to be compassionate or kind. Enablers included building compassion into organisational policies, processes, practices and activities, compassion contagion, training of staff, leading with compassion, and kind and compassionate communication. Outcomes of kindness included building positive and healthy relationships with stakeholders, supporting positive experiences among stakeholders, and contributing to an organisation's profitability, productivity, performance and standing in the community.

In the wake of Australia's Robodebt policy scandal and public critiques about poor practices in other social care bureaucracies such as the NDIS, our scoping review revealed that organisational kindness and compassion is possible and can lead to positive outcomes for those working in organisations and their stakeholders. Opposing misperceptions of kindness as weakness, we recognise the strength shown by politicians and organisational leaders to not become frozen by the personal risks involved in openly supporting kind and compassionate policies, practices, and processes. Strong organisational leaders are prepared to openly speak out against unkindness and to retain their focus on achieving positive outcomes for external clients and stakeholders and implement kindness and compassion within the organisations they lead.

Reflecting the exploratory nature of our study, we highlight the need for future studies that investigate the advancement of compassion and kindness towards external organisational stakeholders. Possible avenues for future research include an investigation of the role of leadership training to create compassionate and kind organisations, qualitative analysis of the impacts of kindness on external stakeholders, and assessment and comparison of different organisational structures and their respective capacities to promote compassion among bureaucratic leaders. In particular, we recognise the importance of kindness and compassion towards external stakeholders in social welfare and the need for future research to focus on this sector in moving beyond education and healthcare sectors. Broad primary research in a wide range of sectors, such as through surveys, interviews, or case studies with external stakeholders, could assist to advance a comprehensive policy understanding of organisational compassion and kindness.

## Author contributions

**Conceptualization:** Jennifer Smith-Merry, Nicola Hancock.

**Data curation:** Jennifer Smith-Merry.

**Formal analysis:** Jennifer Smith-Merry, Damian Mellifont, Justin Newton Scanlan.

**Investigation:** Jennifer Smith-Merry.

**Methodology:** Jennifer Smith-Merry, Justin Newton Scanlan, Nicola Hancock.

**Writing – original draft:** Jennifer Smith-Merry, Damian Mellifont.

**Writing – review & editing:** Jennifer Smith-Merry, Damian Mellifont, Justin Newton Scanlan, Nicola Hancock.

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
