## [Decision Letter · Decision Letter 0]

20 Aug 2024

PONE-D-24-19927Organisational kindness and compassion: what are the barriers, enablers and outcomes for clients and stakeholders?PLOS ONE?

Dear Dr. Mellifont,

We look forward to receiving your revised manuscript.

Kind regards,

Iskra Alexandra Nola

Academic Editor

PLOS ONE

Journal Requirements:

3. Please ensure that you include a title page within your main document. You should list all authors and all affiliations as per our author instructions and clearly indicate the corresponding author.

4. Please ensure that you include a title page within your main document. We do appreciate that you have a title page document uploaded as a separate file, however, as per our author guidelines (http://journals.plos.org/plosone/s/submission-guidelines#loc-title-page) we do require this to be part of the manuscript file itself and not uploaded separately.

Reviewers' comments:

Reviewer's Responses to Questions

**Comments to the Author**

1. Is the manuscript technically sound, and do the data support the conclusions?

Reviewer #1: Partly

Reviewer #2: Partly

2. Has the statistical analysis been performed appropriately and rigorously?

Reviewer #1: N/A

Reviewer #2: I Don't Know

3. Have the authors made all data underlying the findings in their manuscript fully available?

Reviewer #1: No

Reviewer #2: Yes

4. Is the manuscript presented in an intelligible fashion and written in standard English?

Reviewer #1: Yes

Reviewer #2: Yes

Reviewer #1: Title: Organisational kindness and compassion: What are the barriers, enablers and outcomes for clients and stakeholders?

Authors: Jennifer Smith-Mey, Damian Mellifont, Justin Newton Scanlan, and Nicola Hancock

Manuscript Number: PONE-D-24-19927

Overview: This is a “scoping review” paper on the constraints and prospects of organizational kindness and compassion for external beneficiaries. It utilizes a three-stage approach to “scoping” the existing literature. It finds that there are indeed obstacles as well as enablers of organizational kindness and compassion. The outcomes of kindness and compassion can be positive or negative depending on how the barriers and enablers are managed. The paper concludes organizational kindness and compassion “is essential for ongoing trust in health and social care institutions and government policy.” Hence, it recommends that institutions pursue kindness and compassion as deliberate policies.

The paper deals with a topic of great importance. While it is a great project, below I describe areas that would benefit from additional consideration.

Abstract: The abstract is well done but it appears to be too long. I also stress that the abstract should summarize the introduction.

Introduction: First, the introduction should summarize the whole paper by (a) stating the problem to be addressed and justifying the value of doing so, (b) specifying how the problem is or was addressed (materials and methods), (c) outlining the results and their implications, and (d) drawing a conclusion and what it recommends for policy and further research. Some elements of these are there but are not all expressed succinctly, and they should be.

Second, since there is no formal literature review, it would be informative if the authors stated what is missing from the existing literature and what this paper contributes to what is already known. In the Paragraph starting with “Stepping back …” (p. 4), there is a need to provide references specific to “organizational kindness and compassion.”

Third, on p. 5, “kindness” is not well characterized. For example, is kindness a direct or incidental function of policy? How is kindness and compassion measured? In private (for-profit) organizations, do “external clients and stakeholders” include investors or not?

Method: This section appears well-done. However, I wished each stage was better motivated. How did the authors determine “potentially relevant studies” and how many of these did the scoping turn up? Stage 4 (p. 8) talks about “a charting framework” what is that (describe) and how was it “developed”? There are also undefined acronyms like DM, JSM, JNS, and NVivo – what are these, and why should the reader accept the soundness of all these?

Results: The results are reasonable. However, occasionally, the paper appears not to distinguish clearly enough the difference between organizational kindness and compassion and the kind and compassionate actions of individuals. If Microsoft CEO donates to an orphanage in Mozambique, is he or Microsoft who is kind and compassionate?

Discussion: I expected more examples than just education and health care to minimize the fallacy of composition.

Limitations: Well-taken.

Conclusion: This one is the strongest section of the paper in terms of presentation. The only question it raises is why the conclusion does not have any implications for future research. Should the reader conclude this case is open and shut?

General: This is a great project and I enjoyed reading it – good luck!

Reviewer #2: The paper explores the impact of organizational kindness on stakeholders, focusing on the social welfare area. Moving from the Robodebt policy scandal in Australia and its repercussions on participants, the authors conduct a scoping review on barriers, enablers, and outcomes of organizational kindness and compassion for clients and stakeholders. While this study addresses a critical issue, there are several areas where the paper could be improved to enhance its rigor, clarity, and overall impact.

Strengths:

• The topic is highly relevant, especially in light of recent scandals like Robodebt, which underscores the need for organizational kindness.

• The method is well-described, providing a clear framework for how the scoping review was conducted.

Areas for Improvement:

1. Support for Claims:

o The claims regarding the impact of kindness and the listed factors need to be better supported by data. The extent to which these factors hamper or facilitate kindness and their impact on stakeholders should be rigorously displayed. Quantitative analysis from the literature review should be presented more clearly to substantiate these findings.

2. Scope of Literature:

o The available literature is limited and mainly focused on healthcare and education. There appears to be a gap in sources specifically addressing the welfare area. This limitation should be explicitly acknowledged in relevant sections of the paper.

3. Content Revision and Organization:

o The remaining paragraphs require content revision and re-organization to enhance readability. Topics should be consolidated rather than scattered across different paragraphs.

4. Relevance and Completeness:

o It might be worth mentioning the reforms prompted by the negative consequences of the Robodebt scandal in the management of welfare systems and the implementation of automated decisions in recent years, for better completeness.

Detailed Recommendations:

1. Theoretical Frameworks:

o Apply theoretical frameworks from organizational behavior, compassion, and social welfare to structure analysis and draw meaningful conclusions.

2. Cross-Sector Analysis:

o Draw parallels from existing research in healthcare and education, explaining how findings might be relevant or transferable to the welfare sector.

3. Future Research:

o Emphasize the need for future research focused specifically on organizational kindness in the welfare sector, calling for further studies by other researchers and policymakers. This could include broadening the literature review scope to include related fields such as social work, public policy, and community health, which may offer insights applicable to the welfare context. Conduct primary research within the welfare area, including surveys, interviews, or case studies with stakeholders, to gather specific data on organizational kindness, for a comprehensive understanding of organizational kindness in the welfare sector.

Specific Structural and Content Recommendations:

1. Abstract:

o Mention the Australian policy (Robodebt) given its relevance to the research. This will provide better context and highlight the significance of the study from the outset.

2. Introduction:

o Move the New Zealand example to the introduction and reference it when discussing barriers instead of waiting for the discussion section to introduce it. This helps to set the stage and provides a comparative context early in the paper.

3. Conceptual Clarity:

o Distinguish between kindness and compassion: while connected, they are distinct concepts. Consider listing compassion among the enablers. This would also necessitate a revision of the research title.

o Provide detailed definitions of key terms such as "inauthentic behaviors," "respect," and "professional behavior", acknowledging their potential for varied interpretations.

4. Data Presentation:

o Enhance the presentation of quantitative findings. Clearly display how factors identified from the literature review impact organizational kindness and the resulting effects on stakeholders.

5. Revisions for Readability:

o Reorganize content to ensure each topic is discussed in one place. Avoid scattering related information across different paragraphs.

o Ensure a logical flow of ideas from the introduction through to the conclusion.

6. Discussion:

o Currently, the discussion is scattered, which hampers the reader's ability to get an overview of the key points.Use this section to interpret the results, linking them back to the research questions and existing literature.

o Discuss the practical implications for organizational policy and practice, especially in light of the Robodebt scandal.

o Provide clear recommendations based on the results.

o Identify areas for future research to address gaps and build on the findings of the current study.

7. Policy Reforms:

o Mention the reforms prompted by the negative consequences of the Robodebt scandal in the discussion. This will provide a more comprehensive view of the changes in welfare systems and the management of automated decisions.

8. Conclusion:

o Summarize the key findings and their implications succinctly.

o Reinforce the importance of organizational kindness and compassion in the context of social welfare.

o Highlight the need for further research to explore these concepts more deeply in other sectors beyond healthcare and education.

By addressing these areas, the paper will be better positioned to make a significant contribution to the understanding of organizational kindness and its impact on stakeholders, particularly within the social welfare sector. The suggested improvements aim to enhance the clarity, rigor, and overall coherence of the study, ensuring that the findings are robust and actionable.

**Do you want your identity to be public for this peer review?** For information about this choice, including consent withdrawal, please see our Privacy Policy

Reviewer #1: **Yes: ** Voxi Heinrich Amavilah

Reviewer #2: **Yes: ** Sara Giorgi

---

## [Author Response · Author response to Decision Letter 1]

4 Oct 2024

Please see our response to reviewers file as per attached.

---

## [Editor Report · Decision Letter 1]

17 Oct 2024

PONE-D-24-19927R1

Organisational kindness and compassion: what are the barriers, enablers and outcomes for clients and stakeholders?

PLOS ONE

Dear Dr. Mellifont,

Thank you for submitting your manuscript to  PLOS ONE., and for responding to our recent requests regarding your submission. Unfortunately, in our final editorial checks of the documents that you supplied, we have concluded that your submission does not comply with our policies around data availability despite our repeated requests. We are therefore overturning the provisional editorial accept decision, and rejecting this manuscript.  

PLOS journals require authors to make all data necessary to replicate their study’s findings publicly available without restriction at the time of publication (https://journals.plos.org/plosone/s/data-availability). In this case, the following underlying data were not provided as requested: 

-A numbered table of all studies identified in the literature search, including those that were excluded from the analyses.  

As a result of these concerns, we cannot consider the manuscript for publication. I am very sorry that this issue was identified at such a late stage.  

Kind regards,

Vanessa Carels

Staff Editor

PLOS ONE

**Additional Editor Comments:**

Dear dr Mellifont,

Thank you for your revised version of paper. Please consider to rewrite following sentences to give them more clarity:

1. Of these we identified a total of 25 publications that met our inclusion (see Figure 1 for PRISMA diagram). - did you mean criteria?

2. Reflecting upon strong criticisms received by New Zealand’s former Prime Minister Jacinda Ardern (61), we note, however, that these kinds of negative and unjustified stakeholder

responses to compassionate leadership are not a given. - not understandable

3. The findings were mainly limited to qualitative studies, and impacted by small sample sizes. And the possibility of selection bias with study participants not randomly selected. - maybe not to start the second sentence with And?

4. These results provide a frame for considering the results of organisational compassion and kindness on external stakeholders, and the factors that limit or make this possible. - I am not sure what you meant by this sentence.

Thank you,

Kind regards,

Iskra A. Nola

- - - - -

---

## [Author Response · Author response to Decision Letter 2]

4 Dec 2024

Please see the uploaded rebuttal letter (Response to Reviewers – Track changes02122024) and the Additional Editor Comments and Authors’ Responses files.

---

## [Editor Report · Decision Letter 2]

6 Jan 2025

PONE-D-24-19927R2Organisational kindness and compassion: what are the barriers, enablers and outcomes for clients and stakeholders?PLOS ONE?

Dear Dr. Mellifont,

Thank you for submitting your manuscript to PLOS ONE. After careful consideration, we feel that it has merit but does not fully meet PLOS ONE’s publication criteria as it currently stands. Therefore, we invite you to submit a revised version of the manuscript that addresses the points raised during the review process.

**ACADEMIC EDITOR:**

I appreciate the substantial effort you have made to address the concerns raised in the previous review. The manuscript demonstrates strong potential for contribution to the field and aligns with PLOS ONE’s scope.

However, upon careful evaluation of the revised submission, I have identified a few areas that require further refinement to ensure clarity, rigor, and alignment with the journal's standards. These refinements will improve the manuscript's readability and strengthen its suitability for peer review. Below, I outline the specific concerns, with examples, to guide your revisions:

Clarity and Readability

Certain sentences and sections remain unclear, affecting the readability and comprehension of the manuscript. Please revise these to ensure the meaning is precise and accessible to a broad audience.

Example 1: "Reflecting upon strong criticisms received by New Zealand’s former Prime Minister Jacinda Ardern (61), we note, however, that these kinds of negative and unjustified stakeholder responses to compassionate leadership are not a given."

Suggested revision: Simplify the phrasing to improve clarity. For instance: "Jacinda Ardern’s leadership faced criticism, but negative responses to compassionate leadership are not inevitable."

Example 2: "These results provide a frame for considering the results of organisational compassion and kindness on external stakeholders, and the factors that limit or make this possible."

Suggested revision: Rephrase to specify what the "frame" entails and the implications. For instance: "These findings offer a framework for understanding how organizational compassion and kindness influence external stakeholders, as well as the factors that enable or hinder these practices."

Focus on External Stakeholders

While the manuscript addresses kindness and compassion, the examples and discussions are heavily focused on healthcare and education. Broaden the discussion to other sectors or justify the emphasis on these fields.

Add a statement to the discussion section explaining why healthcare and education dominate the findings or include insights from additional sectors, if possible.

Discussion of Limitations

The limitations section could benefit from further elaboration, particularly on issues such as selection bias, reliance on qualitative data, and the potential for small sample sizes in the included studies.

Avoid starting a sentence with "And" and expand on the implications of these limitations for generalizability. For instance: "The findings are primarily based on qualitative studies with small sample sizes, raising concerns about generalizability. Additionally, the possibility of selection bias, due to non-random participant selection, should be acknowledged."

Consistency in Terminology and Style

Ensure consistency in terminology and avoid stylistic issues, such as informal sentence structures or overly complex phrasing.

Example: Avoid phrases like "we note, however, that…" or starting sentences with conjunctions like "And." These can be revised for a more formal and polished tone.

Grammatical and Typographical Issues

Conduct a final proofread to address minor grammatical errors and typographical issues.

References and Citations

Some references, particularly URLs and book references, may require adjustments to align with the journal's format.

We look forward to receiving your revised manuscript.

Kind regards,

Mohd Ismail Ibrahim, MCom.Med

Academic Editor

PLOS ONE
---

## [Decision Letter · Decision Letter 3]

22 Apr 2025

PONE-D-24-19927R3Organisational kindness and compassion: what are the barriers, enablers and outcomes for clients and stakeholders?PLOS ONE?

Dear Dr. Mellifont,

Thank you for submitting your manuscript to PLOS ONE. After careful consideration, we feel that it has merit but does not fully meet PLOS ONE’s publication criteria as it currently stands. Therefore, we invite you to submit a revised version of the manuscript that addresses the points raised during the review process.

We look forward to receiving your revised manuscript.

Kind regards,

Mohd Ismail Ibrahim, MCom.Med

Academic Editor

PLOS ONE

Journal Requirements:

Additional Editor Comments:

***Comments from the editorial office:* Upon internal evaluation of the reviews provided, we kindly request you to disregard the reviewer report provided by Reviewer 4. No amendments are required in response to reviewer 4’s comments.**

Reviewers' comments:

Reviewer's Responses to Questions

**Comments to the Author**

Reviewer #3: All comments have been addressed

Reviewer #4: All comments have been addressed

2. Is the manuscript technically sound, and do the data support the conclusions?

Reviewer #3: Yes

Reviewer #4: Yes

3. Has the statistical analysis been performed appropriately and rigorously?

Reviewer #3: N/A

Reviewer #4: Yes

4. Have the authors made all data underlying the findings in their manuscript fully available?

Reviewer #3: Yes

Reviewer #4: Yes

5. Is the manuscript presented in an intelligible fashion and written in standard English?

Reviewer #3: Yes

Reviewer #4: Yes

Reviewer #3: This paper covers the very interesting and unstudied organizational phenomenon of kindness. It is clearly written, methods are sound and have fidelity to Arksey & O'Malley, and results well-constructed. The discussion talks back to what is known about the organizational kindness, and limitations are sound. Overall, this is a good paper and engages with the very current political zeitgeist (at least in the U.S.) on empathy as damaging, rest as weakness, and compassion as problematic.

That said, I have a few comments to the authors:

- I am uncomfortable with the reification (see Oxford Reference entry) -- namely that an organization can be kind or express kindness. Rather, it is more accurate to say that it is the people who work there -- at all levels of decision-making and power -- who behave in kind ways and/or follow policies and procedures that direct them to be kind? It is the policies that have been produced by people that enact/constrain kindness. I think as long as the authors write a sentence to acknowledge that they are reifying organizations (and even argue in favor of doing so), it will read as a more scholarly paper.

- has it been established what the examples, typologies, themes, etc. of organizational kindness are? this almost seems to be a first step in the authors' research, but is missing from the paper.

- would suggest that the authors explain the nuanced differences/similarities between kindness and compassion that make these terms interchangeable in the study

- suggest a table with the themes, and perhaps a concept map or diagram? how do they relate with each other?

Reviewer #4: Thank you for the opportunity to review this manuscript. It explores the organisational dimensions of kindness and compassion in relation to external stakeholders — an underexamined yet crucial subject in public administration, human services, and organisational theory. This is an insightful and well-constructed article with timely relevance.

The manuscript addresses a highly relevant and timely topic in organisational research. The scoping review methodology employed is well-suited to exploring the existing literature, and the authors have successfully identified the key barriers, enablers, and outcomes associated with organisational kindness and compassion, while proposing a thoughtful set of future research directions.

The study’s focus on real-world examples, such as the Australian Robodebt policy, adds practical depth to the theoretical discussions. The manuscript contributes significantly to the literature by examining an under-researched area and demonstrating the importance of kindness and compassion in building trust-based relationships between organisations and external stakeholders, particularly in social welfare contexts.

Strengths of the Manuscript:

• The paper provides a comprehensive and well-structured scoping review grounded in Arksey and O'Malley's framework. Inclusion and exclusion criteria are clearly defined and consistently applied.

• The thematic analysis is effectively organised into three domains — barriers, enablers, and outcomes — which are well supported by literature.

• The discussion connects theory and practice, with relevant policy examples (such as Robodebt) reinforcing the real-world significance of the findings.

• The focus on organisational structures, not just individual behaviour, contributes meaningfully to ongoing debates about ethical and relational bureaucracies.

Suggestions for Improvement:

1. Broaden Sectoral and Geographic Scope:

The current sample of studies is heavily concentrated in healthcare and education sectors from high-income, English-speaking countries. The authors should consider expanding the scope or, at a minimum, explicitly acknowledge this bias and its implications for the generalisability of findings. Studies from other sectors such as business, government services (outside healthcare/education), or civil society organisations could enrich the analysis.

2. Add More Real-World Case Studies:

While the Robodebt example is valuable, incorporating additional case illustrations from different sectors (e.g., social housing, refugee services, community organisations) would enhance the practical insights. These examples could also showcase the diversity of ways in which compassion is operationalised.

3. Clarify Use of NVivo and Thematic Analysis:

The paper briefly mentions NVivo software but lacks detail on how it was employed. A short description of the coding process, theme development, and any steps taken to ensure analytical rigour (e.g., double coding, audit trail) would increase transparency and strengthen trust in the findings.

4. Address Generalisability and Methodological Limitations:

The findings rely mostly on small-scale qualitative studies. This limitation is acknowledged, but could be expanded to include suggestions for future research designs (e.g., mixed-method approaches, longitudinal or cross-national comparisons) that would help validate and expand on the themes identified.

5. Deepen the Conceptual Distinction Between Kindness and Compassion:

The paper touches on definitional issues early on but could further clarify how these concepts are differentiated across sectors and disciplines. A comparative table or conceptual framework could help readers navigate the nuances, particularly for an interdisciplinary audience.

6. Improve Outcomes Synthesis:

The outcomes identified are relevant but currently presented more as a list. A conceptual framework or logic model showing how enablers and barriers influence outcomes at emotional, behavioural, and organisational levels would make this section more analytical and policy relevant.

7. Enrich Future Research Directions:

The suggestions for further research are appropriate but could be more specific. For instance, future studies could examine: (a) Long-term impacts of compassionate leadership on staff retention and morale, (b) Organisational metrics for measuring compassionate practices, (c) Cross-cultural comparisons of compassion practices in public services

8. Language and Readability:

The manuscript is generally well written, clear, and accessible. However, there are several instances of redundancy and repetitive phrasing that, if revised, would improve conciseness and flow:

• The phrase “kindness and compassion” is often repeated in proximity within the same sentence or paragraph. For example, in one sentence the authors write: “Organisational kindness and compassion have been linked to improved outcomes for clients and service users, where kindness and compassion from staff and services are experienced as a core component of quality care and support.”

This repetition can be reduced for clarity: “Organisational kindness and compassion are recognised as core components of quality care and support, contributing to improved outcomes for clients and service users.”

• Redundant phrasing is also evident in the methodology section, where the authors state:

“...after a rigorous screening and selection process, which followed the scoping review methodology.”

Since methodological rigour is already implied in the framework used, this sentence could be shortened to: “...after a screening process following the scoping review methodology.”

• Conceptual repetition also appears in the Barriers section, for example:

“Bureaucratic rules and system demands were often in conflict with the ability to act compassionately. These organisational demands and pressures often restricted or prevented acts of kindness or compassion.”

This could be condensed: “Bureaucratic systems and organisational pressures often restricted staff from acting compassionately.”

• The use of phrases such as “clients and service users” interchangeably and repeatedly in the same sentence should be avoided. Choose one term and use it consistently.

• In the conclusion, ideas are occasionally restated without added insight. For instance:

“...kindness and compassion can be enacted at an organisational level. Organisations can cultivate kindness and compassion structurally...”

could be simplified to: “This review demonstrates that organisations can cultivate kindness and compassion structurally through policy and culture.”

A final round of proofreading is recommended to streamline the prose, eliminate unnecessary repetition, and ensure consistency in terminology.

9. Ethical and Publication Transparency:

No ethical concerns are noted, but it would be appropriate for the authors to state explicitly that the manuscript is original, has not been previously published, and is not under consideration elsewhere. Additionally, the ethical sourcing of data included in the review could be briefly acknowledged.

Conclusion and Recommendation:

This is a highly promising manuscript that contributes significantly to our understanding of how kindness and compassion can be structured and embedded within organisational systems. It is especially timely considering ongoing debates about public service reform and citizen-centred bureaucracy.

With minor revisions to strengthen methodological transparency, broaden empirical context, and enhance conceptual synthesis, this paper will make a valuable and impactful contribution to scholarship and practice.

**Do you want your identity to be public for this peer review?** For information about this choice, including consent withdrawal, please see our Privacy Policy

Reviewer #3: No

Reviewer #4: **Yes: ** Istiana Hermawati

---

## [Author Response · Author response to Decision Letter 4]

30 Apr 2025

As per the attached 'Reponse to Reviewers' file:

1. The acknowledgement sentence has been added as suggested (see page 8).

2. We now map out this more clearly and provide a definition of organisational kindness. We note shared constructions of organisational kindness is a persisting, social and reflective process, but that organisational kindness is also reflected in a range of practical examples within the literature (see changes on pages, 5, 6, 8).

3. The nuances between kindness and compassion that make these terms interchangeable in the study are further explained as suggested (see pages 5, 6, 8).

4. Figure 2 has been added as a concept map (PACE verified) as suggested (pages 12 and 13).

---

## [Decision Letter · Decision Letter 4]

21 May 2025

Organisational kindness and compassion: what are the barriers, enablers and outcomes for clients and stakeholders?

PONE-D-24-19927R4

Dear Dr. Mellifont,

We’re pleased to inform you that your manuscript has been judged scientifically suitable for publication and will be formally accepted for publication once it meets all outstanding technical requirements.

Kind regards,

Mohd Ismail Ibrahim, MCom.Med

Academic Editor

PLOS ONE

Additional Editor Comments (optional):

Reviewers' comments:

Reviewer's Responses to Questions

**Comments to the Author**

Reviewer #3: All comments have been addressed

2. Is the manuscript technically sound, and do the data support the conclusions?

Reviewer #3: Yes

3. Has the statistical analysis been performed appropriately and rigorously?

Reviewer #3: Yes

4. Have the authors made all data underlying the findings in their manuscript fully available?

Reviewer #3: Yes

5. Is the manuscript presented in an intelligible fashion and written in standard English?

Reviewer #3: Yes

Reviewer #3: The authors have adequately addressed my concerns and provided very helpful figures to the manuscript that emphasize their findings.

**Do you want your identity to be public for this peer review?** For information about this choice, including consent withdrawal, please see our Privacy Policy

Reviewer #3: No

---

## [Editor Report · Acceptance letter]

PONE-D-24-19927R4

PLOS ONE

Dear Dr. Mellifont,

I'm pleased to inform you that your manuscript has been deemed suitable for publication in PLOS ONE. Congratulations! Your manuscript is now being handed over to our production team.

Kind regards,

on behalf of

Dr. Mohd Ismail Ibrahim

Academic Editor

PLOS ONE